# Designing Flexible-Bus System with Ad-Hoc Service Using Travel-Demand Clustering

**Xuekai Cen, Kanghui Ren, Yiying Cai * and Qun Chen**

School of Traffic and Transportation Engineering, Central South University, Changsha 410075, China
* Correspondence: caiyiying@csu.edu.cn

**Abstract:** Providing direct and affordable transit services for travelers is the goal of the evolving flexible-bus (FB) system. In this study, we design an FB system with an ad-hoc service, to supplement traditional public transit and provide a better FB service. We first build up a mathematical model to optimize bus-stop sites, routes, and schedules, where the unmet travel demand is served by an ad-hoc service with relatively high cost. Then, we cluster travel demand spatially and temporarily, using the ST-DBSCAN algorithm. We use the simulated-annealing algorithm, which has better convergence and diversity than other heuristic algorithms, to solve the suggested model in large-scale networks. To demonstrate the effectiveness of the proposed model, we run experiments on a small network and a large real-world network of Shenzhen airport, which shows that the FB system with ad-hoc service can reduce overall cost and improve social welfare, compared to taxies and FB only. In addition, it provides affordable transit services with shorter walking distances and lower waiting times, which can be deployed in airports or high-speed railway stations with massive, irregular travel demands.

**Keywords:** flexible-bus system; travel-demand clustering; ad-hoc service

**MSC:** 90B06

## 1. Introduction

Traditionally, without real-time demand information, public transport (PT) services are operated on fixed routes with fixed schedules (FRFS), regardless of the actual demand, which is neither efficient nor cost-effective. With the emergence of information technology, China's urban-traffic travel structure has changed dramatically. Although flexible, demand-responsive services such as taxis are too expensive to function as a significant PT option, due to a lack of economies of scale [1].

As a supplement to traditional public transportation, demand-responsive transit (DRT) is gaining increasing amounts of interest from the transportation industries [2]. In particular, a flexible bus (FB), with flexible stops and departure times, can accommodate customers' varied travel demands while maintaining the basic service charge, and passengers can benefit from reservation services.

It is time-consuming, ineffective, and expensive for current FB systems to manually plan FB lines by compiling travel information from online surveys [3]. Compared to traditional transit-network design, establishing bus routes for FB systems presents some unique challenges: (i) Effectively identifying the desired travel-demand patterns. In order to be economical, FB systems should arrange their transit resources to fit with the huge travel demands with proximate origins and destinations and similar departure times. (ii) To achieve a balance between public services and market rewards, the FB path should be founded on scientific research and practical planning [4,5].

In this study, we use the ST-DBSCAN method to handle taxi GPS trajectories that can uncover potential ride-sharing needs to cut down on wasteful bus-stop durations in the FB-routing problem. Then, in order to achieve the lowest possible overall cost for the bus company and its customers, we develop a flexible-bus (FB) model with ad-hoc

service, which may also be thought of as the cost of unserved demand. That is, an ad-hoc service will be used to fulfill some demands when the regular fleet is unable to do so due to vehicle capacity or travel-time restrictions. In comparison to existing heuristic methods, we use the simulated-annealing hermetic algorithm to solve the suggested model in large-scale networks, which has greater convergence and diversity. Last but not least, the formulation and solution algorithms are applied to small-scale instances and the real data in Shenzhen, China, to prove their applicability. We show in the case study that, in terms of overall travel costs and social welfare, the FB system outperforms the taxi service and traditional bus service. In all, it has two benefits: (i) The clustering algorithm can effectively integrate passengers' demand with similar spatiotemporal characteristics, considering reasonable walking distance and waiting time from the aspect of passenger convenience. (ii) The flexible bus system with an ad-hoc service may successfully lower operating costs and increase customer convenience, which can further maintain balance among different stakeholders.

To sum up, the contributions of the study are as follows.

- A mathematical formulation for the FB routing problem is provided, to simultaneously optimize bus-stop locations, bus routes, and schedules. In order to fulfill some uneconomic travel demands, we add an ad-hoc service that is more accurate with regard to the actual traffic situation and could even further reduce total cost. To solve the suggested model in large-scale networks, we use the simulated-annealing algorithm, which exhibits greater convergence and variety than alternative heuristic algorithms.
- In order to show the adaptability of the clustering approach and FB model, we set a case study by using the taxi-trajectory data from Shenzhen, China. The findings demonstrate that our model can produce flexible bus routes at a lower overall cost, compared with the traditional bus route and taxi service. Furthermore, these FB routes are capable of providing efficient transit services in terms of less walking distances and lower travel costs.

The remainder of this paper is structured as follows. The background of FB and related work is highlighted in Section 2. The mathematical framework of flexible-bus-line planning is developed in Section 3. The clustering algorithm and heuristic-solution framework are described in Section 4. The numerical experiment and the case study for assessing the performance of the flexible-bus system are presented in Sections 5 and 6, respectively. In Section 7, we conclude this paper and give possible extensions of future research.

## 2. Literature Review

### 2.1. Traditional Transit-Network Design

By optimizing certain target functions such as maximum service coverage and minimum passenger discomfort, transit-network design (TND) [6–8] establishes bus-route layouts and the associated features such as bus frequencies [9], timetables [10,11], and space between stops [5].

The majority of TND solutions optimize route designs based on a predetermined road-network infrastructure. Some studies assumed that the bus stops had been placed strategically on the road network [12–15]. Other researchers generated bus routes as sequences of nearby nodes of the road network [16,17]. Several studies simultaneously improved the routes and placements of bus stops [18,19]. In these studies, a list of potential stops was taken for granted, making it simple to incorporate the stop-site-choice problem into routing decisions. In fact, bus stops are crucial parts of FB systems, which control how easily buses are routed. Therefore, some study papers have steadily investigated how to determine the positions of bus stops. For instance, Ceder et al. [20] calculated the distance between bus stops along an existing route. In addition, Ma et al. [21] clustered origin and destination locations, independently. Using the connection between bus-service routes and bus-station sites, Crainic et al. [22] solved the issue of passenger assignment and station-location selection. Yu et al. [23] suggested using the web to create tiered service regions and stops for individualized transit networks.

## 2.2. Travel-Demand Clustering

Large-scale spatiotemporal data, such as mobile-phone data, smart-card data from public transportation, taxi-GPS data, etc., have emerged with the growth of information technology, and have been extensively employed in studying human-movement patterns. Numerous academics work to effectively convey the vast and extensive movement of people and to identify regional travel patterns and regularities on an urban scale [24]. The advent of the sharing economy has garnered a lot of attention in recent years, as well. Sharing transportation services can dramatically reduce traffic congestion [25], make first- and last-mile connections to public transportation easier [26], and increase a city's social, economic, and environmental sustainability [27].

Lyu et al. [5] offered a productive grid-density-based clustering algorithm to identify probable commuter groups with comparable demands based on their origins, destinations, and departure timings. Based on the aforementioned investigations, we apply the spatial-temporary DBSCAN algorithm to group users' demands according to comparable spatiotemporal properties, producing a collection of clusters and accessible stops. The deployed stops may offer better accessibility for FB clients, due to the reasonable walking distances and waiting times.

## 2.3. Flexible-Bus Network Design

The FB has become more popular in China recently as a demand-responsive transit mode [3]. It offers passengers a user-focused, transfer-free, and cost-effective transportation service by grouping similar travel requests [28]. Tao et al. [3] were the first to comprehensively propose the service-design method of FB. The procedure begins with collecting passenger subscriptions, followed by route planning and vehicle dispatching, and then employs real-time control strategies to modify travel schedules in response to the current situation.

Some studies have concentrated on FB-system design and optimization [9,29–31]. Given the fixed FB lines and travel demands, Cao et al. [32] investigated the passenger-assignment problem for FB networks. In order to avoid traffic jams, Zhou et al. [33] examined the FB bus-dispatching problem and developed a diversion method and a vehicle-replacement approach. Ma et al. [21] suggested a paradigm for immune-genetic-halt planning and scheduling for FB systems. Following that, Lyu et al. [5] created a comprehensive framework for FB planning that takes demand clustering, station layout, route planning, and schedule formulation into account. In order to reduce operating costs and passenger costs, Chen et al. [34] devised a similar integrated strategy that takes into account walking distances between passengers and bus stops. The FB-routing problem was added to with time-dependent trip duration and path selection by Guo et al. [35]. Moreover, Wu et al. [36] proposed and evaluate a customized FB service for railway stations and airports in China, where they took account of a routing problem with a time-dependent travel time and late customers.

Therefore, we propose an FB system with an ad-hoc service that could result in lower overall costs and be more appropriate for real-world application. Due to vehicle-capacity issues or time restrictions associated with detours, certain requests may not be met by the normal fleet. In these cases, an ad-hoc service will be used to fill the gap left by the unmet demand.

## 2.4. Solution Algorithm

Since the FB-optimization problem is combinatorial and non-deterministic polynomial-time hard (NP-hard), it has been thought to be hard to find the exact solution in a reasonable amount of time. Numerous academics have created numerous algorithms to improve the approach taken to address the issue of a flexible public-transportation system.

The genetic algorithm (GA) is a stochastic search technique that uses the biological world's survival of the fittest and other natural principles to locate the best solution space. It is hopeful that GA can optimize complex mathematical models such as the vehicle-routing

problem (VRP) [37–41], without the need to calculate the objective function's derivatives. To find the best option, GA has been used in numerous studies. A routing issue for an advanced public-transportation system was resolved by Uchimura et al. [42]. Transit routes and frequencies were optimized by Shrivastava et al. [43]. However, traditional GA encounters various difficulties when looking for the answer (for example, premature or local optimal solutions [44]). Some studies combined GA with elements of other algorithms to improve performance, in order to address this issue. Masmoudi et al. [38] developed a hybrid GA with a local-search method.

Furthermore, the application of particle swarm optimization (PSO) to VRP problems exhibits great performance in continuous problems [44–46]. Goldbarg et al. [47] created a multi-agent hybrid-element heuristic algorithm within the PSO framework. Zhou et al. [48] used a particle-swarm-optimization technique. Although the particle-swarm-optimization algorithm performs well in the early stages of the search, it is simple to enter the local optimal but impossible to exit it in the latter stages of optimization exploration, which causes the convergence to become sluggish or even to stop.

We use the simulated-annealing (SA) algorithm to solve the issue in the model because the local-search strategy of the SA is more reliable for locating high-quality solutions in VRP problems [49]. Using a temperature-changing schedule, the SA algorithm employs a probabilistic-acceptance technique that involves numerous iteration phases to obtain the global optimum [50]. Because SA even accepts inferior responses with a certain probability, it has an advantage over other comparable metaheuristics in that it may quickly leap into the solution space to find a global-optimum solution and escape from the local minima [51].

## 3. Methodology

### 3.1. Problem Description and Assumption

The flexible-bus-line planning problem is formulated in this part by simultaneously optimizing bus-stop sites, bus routes, and timetables. To find optimal answers, numerical experiments can be utilized to construct the flexible-bus-routing problem with time windows as a mixed-integer problem [52,53]. On a complete graph $G = (V, A)$, the flexible-bus-route design issue is described, where $V = (0, 1, \cdots, N, N + 1)$ is a collection of stations and depots. Since vertex 0 and vertex $N + 1$ represent the same depot, each vehicle should leave from vertex 0 and return to vertex $N + 1$. Let $S = \{1, \cdots, N\}$ represent the collection of stations that have $S = O \cup D$, where $O$ and $D$ stand for the origin and destination sets, respectively. An origin, $r \in O$, and a destination, $s \in D$, are connected to a positive travel demand, $q_{rs}$. Every component of the set $A = \{(i, j) | i, j \in V, i \neq j\}$ connects two vertices in $V$ and is linked with two non-negative values, namely travel time, $t_{ij}$, and distance, $d_{ij}$.

At depot 0, a collection of uniform cars with the highest capacity of $Cap$ is built up. Within predetermined times, vehicles are directed to pick up and drop off passengers before heading back to the depot. The vehicle must enter the station after $ArrT_i$ and exit the station before $DepT_i$, according to the time window $[ArrT_i, DepT_i]$ that is present for each vertex in this situation. Additionally, the service time for each vertex ($i \in V$) is $t_i$, which states the time spent picking up and dropping off passengers. In addition, we set $t_0$ and $t_{N+1}$ equal to zero to prevent wasting time at the depot without picking up or sending passengers. Keep in mind that the arrival time and service time, or $T_i^k + t_i$, are used to represent the vehicle departure time from each station, whereas $t_0 = 0$ is used to represent the vehicle departure time from the depot.

Some suppositions are made in advance, to simplify the problem: (1) the demand for travel between any origin and its corresponding destination is known; (2) the time windows and service times of each vertex are pre-given; (3) the travel time and distance of any arc are pre-given; (4) the capacity and average speed of vehicles are constant and given; and (5) the flexible-bus-transit network is a two-way physical network.

### 3.2. Mathematical Modeling

The notations used throughout this work are presented in Table 1 to facilitate the statement and aid readers in better understanding the context.

**Table 1.** Notation.

| Notation | Definition |
|---|---|
| **Sets and Indices** | |
| $0, N+1$ | Depot instance |
| $V$ | Set of vertices |
| $S$ | Set of stations |
| $i, j$ | Indices of vertices |
| $O$ | Set of origins |
| $D$ | Set of destinations |
| $r$ | Index of origins |
| $s$ | Index of destinations |
| $OD$ | Set of distribution paths(say $r$, $s$), which are sent from the origin, $r \in O$, to the destination, $s \in D$. |
| $A$ | Set of arcs connecting pairs of vertices |
| $(i, j)$ | Index of arcs |
| $\Delta_i^+$ | Set of arcs departing from vertex $i$ |
| $\Delta_j^-$ | Set of arcs returning to vertex $j$ |
| $K$ | Set of vehicles |
| $k$ | Index of vehicles |
| Parameters | |
| $s_{bx}$ | Fixed cost per vehicle (FB) |
| $s_{tx}$ | Fixed cost per taxi (ad-hoc service) |
| $s_{d\_bx}$ | Operating cost of vehicles (FB) per kilometer |
| $s_{d\_tx}$ | Operating cost of taxis per kilometer (ad-hoc service) |
| $s_r$ | Travel-time cost per passenger per minute |
| $s_w$ | Waiting-time cost per passenger per minute |
| $d_{ij}$ | Distance of arc $(i, j)$ |
| $d_{rs}$ | Total distance of an OD pair (say $r$, $s$), which starts from and returns to the depot |
| $t_{ij}$ | Travel time of arc $(i, j)$ by vehicles (FB), also calculated by $t_{ij} = d_{ij}/v_{bx}$ |
| $t_{rs\_tx}$ | Travel time of arc $(i, j)$ by taxis, also calculated by $t_{rs\_tx} = d_{ij}/v_{tx}$ |
| $t_i$ | Service time at vertex $i$ |
| $q_{rs}$ | Travel demand from origin r to destination s |
| $ArrT_i$ | Earliest arrival time at station $i$ |
| $DepT_i$ | Latest departure time at station $i$ |
| $ArrT_{N+1}$ | Earliest arrival time at depot $N+1$ |
| $DepT_0$ | Latest departure time at depot 0 |
| $TC_r$ | The latest arrival time of the clustered demands at pick-up point $r$ |
| $v_{bx}$ | Vehicle average speed (FB) |
| $v_{tx}$ | Taxi average speed (ad-hoc service) |
| $\lambda$ | Minimum-load requirement |
| $Cap$ | Vehicle capacity |
| $I_{min}$ | Lower limit of route length |
| $I_{max}$ | Upper limit of route length |
| $M$ | Maximum station quantity for the route |
| $B$ | A large number, $B = 10^7$ |
| Intermediate variable | |
| $p_i^k$ | Integer variable indicating number of passengers in vehicle $k$ at vertex $i$ |
| $T_i^k$ | Continuous variable indicating arrival time of vehicle $k$ at vertex $i$ |

**Table 1.** *Cont.*

| Notation | Definition |
|---|---|
| Decision variables | |
| $x_{i,j}^k$ | Binary variable, equal to 1 if arc$(i, j)$ is on the optimal route of vehicle $k$ |
| $z_{r,s}^k$ | Binary variable, equal to 1 if an *OD* pair (say $r$, $s$) is served by vehicle $k$ |
| $y_{r,s}^k$ | Integer variable indicating passenger quantity-of-travel demand, $q_{rs}$, in vehicle $k$ |

The mathematical model of the flexible-bus-service-design problem is formulated as a mixed-integer programming, as follows:

$$Min : s_{bx} \sum_{k \in K} \sum_{j \in S} x_{0,j}^k + s_{d_b x} \sum_{k \in K} \sum_{i \in V \setminus \{N+1\}} \sum_{j \in V \setminus \{0\}, i \neq j} x_{i,j}^k d_{ij} + s_r \sum_{k \in K} \sum_{r \in O} \sum_{s \in D} q_{rs} \left( T_s^k - T_r^k \right) +$$

$$s_w \sum_{r \in O} \sum_{s \in D} y_{rs}^k \left( T_r^k - TC_r \right) + \sum_{r \in O} \sum_{s \in D} \left( q_{rs} - \sum_{k \in K} y_{r,s}^k \right) \left( s_{tx} + s_{d_b x} d_{rs} + t_{rs_t x} s_r \right) \tag{1}$$

subject to

$$\sum_{k \in K} \sum_{i \in S, i \neq j} x_{ij}^k \geq 0, \ \forall j \in S \tag{2}$$

$$\sum_{i \in S, i \neq j} x_{ij}^k \leq 1, \ \forall k \in K, j \in S \tag{3}$$

$$\sum_{j \in S} x_{ij}^k = \sum_{j \in S} x_{ji}^k = 1, \ \forall i \in \{0, N+1\}, \forall k \in K \tag{4}$$

$$\sum_{i \in V \setminus \{N+1\}, i \neq j} x_{ij}^k = \sum_{i \in V \setminus \{0\}, i \neq j} x_{ji}^k, \ \forall j \in S, k \in K \tag{5}$$

$$y_{rs}^k > 0 \rightarrow \sum_{j \in S} x_{rj}^k = \sum_{i \in S} x_{is}^k = 1, \ \forall r \in O, s \in D, k \in K \tag{6}$$

$$\sum_{r \in O} \sum_{s \in D} y_{rs}^k \geq \lambda, \ \forall k \in K \tag{7}$$

$$\sum_{k \in K} y_{rs}^k \leq q_{rs}, \ \forall r \in O, s \in D \tag{8}$$

$$x_{ij}^k = 1 \rightarrow p_j^k = p_i^k + \sum_{s \in D} y_{js}^k - \sum_{r \in O} y_{rj}^k, \ \forall k \in K \tag{9}$$

$$p_i^k \leq Cap, \ \forall i \in V, k \in K \tag{10}$$

$$p_0^k = p_{N+1}^k = 0, \ \forall k \in K \tag{11}$$

$$T_j^k - T_i^k - B x_{ij}^k \geq t_{ij} + t_i - B, \ \forall k \in K \tag{12}$$

$$y_{rs}^k > 0 \rightarrow T_s^k \geq T_r^k, \ \forall r \in O, s \in D, k \in K \tag{13}$$

$$ArrT_i \sum_{j \in \Delta_i^+} x_{ij}^k \leq T_i^k \leq DepT_i \sum_{j \in \Delta_i^+} x_{ij}^k, \ \forall i \in S, k \in K \tag{14}$$

$$T_0^k \leq DepT_0, \ \forall k \in K \tag{15}$$

$$T_{N+1}^k \geq ArrT_{N+1}, \ \forall k \in K \tag{16}$$

$$z_{rs}^k = 1 \rightarrow y_{rs}^k > 0, \ \forall r \in O, \ s \in D, k \in K \tag{17}$$

$$z_{rs}^k > 0 \rightarrow T_r^k \geq TC_r \quad \forall r \in O, \ s \in D, k \in K \tag{18}$$

$$z_{rs}^k > 0 \rightarrow \sum_{j \in S} x_{rj}^k = \sum_{i \in S} x_{is}^k = 1, \quad \forall r \in O, \ s \in D, k \in K \tag{19}$$

$$I_{min} \leq \sum_{i \in V \setminus \{N+1\}} \sum_{j \in V \setminus \{0\}, i \neq j} x_{ij}^k d_{ij} \leq I_{max}, \quad \forall k \in K \tag{20}$$

$$\sum_{i \in V \setminus \{N+1\}} \sum_{j \in S, i \neq j} x_{ij}^k \leq M, \quad \forall k \in K \tag{21}$$

Equation (1), which consists of five parts, defines the goal of minimizing the overall cost. The first portion totals all of the vehicles' fixed costs; for example, if a vehicle leaves the depot for service, a fixed cost of the vehicles is generated. The second section totals all operating costs for the vehicles, and the third section assesses the cost of transport for the passengers. The fourth component represents the cost of customers' depot wait times, and the last component computes the ad hoc cost for passengers who were not served. In this study, vehicles might not be able to accommodate every passenger; in other words, the flexible-bus system only offers partial service, since "no-show" passengers will be allocated separately to private taxis, the cost of which includes the fixed cost of the taxi, the cost of travel, and the cost of the passenger on the trip.

The multi-vehicle-routing problem's route restrictions are constraints (2) through (5). According to restrictions (2) through (3), a station may neither be visited by vehicles nor visited more than once by the same vehicle. Each vehicle must begin and terminate each leg of its trip at the depot, according to constraint (4). The flow-balance constraint is the fifth constraint. The relationship between the two variables is demonstrated by constraint (6), which states that for the demand of an *OD* pair (say *r*, *s*) to be accepted by a vehicle *k*, *k* must also travel to the vertices *r* and *s*. Operation constraint (7) stipulates that the number of passengers supplied by the assigned vehicle exceeds the minimum-load requirement in order to ensure the modest profit targets for a long-term development of the flexible bus. The number of passenger provided with journey *OD* (say *r*, *s*) is fewer than or equal to the total number of passengers for this demand, according to constraint (8). Constraints (9) through (11) are limitations on capacity. Based on the in-vehicle passengers of the previous station $p_i^k$, as well as pick-up and delivery passengers at the station *j*, constraint (9) calculates the in-vehicle passengers of the station *j*. Constraint (10) ensures that each vehicle's carrying limit is not exceeded. All vehicles departing and getting back to the depot must be unloaded, according to constraint (11).

The constraints of the time window are addressed by (12)–(16). The constraint of vehicle visiting order is given by Equation (12), which states that the vehicle arrival time at vertex *j* must be at least equal to the total of the vehicle arrival time at the preceding vertex *i*, the service time and the journey time from *i*. The visiting sequence of the paired origin and destination is constrained by constraint (13). If a vehicle (say *k*) transports an *OD* pair (say *r*, *s*), the vehicle's arrival time at destination s will be longer than that of origin *r*. The vertices must be reached and left by the vehicle within each specified time interval, subject to constraints (14) through (16). The relationship among the three variables is demonstrated by constraints (17)–(19), which state that for an *OD* pair (say *r*, *s*) to be served by a vehicle *k*, *k* must also travel to the vertices *r* and *s*. Meanwhile the arriving time at *r* of vehicle *k* must be more than the total passenger arrival time at *r*, when the demands are clustered. In the same way, if the service of an *OD* pair (say *r*, *s*) is offered by a vehicle *k*, the loading must be more than 1 passenger.

Lastly, the modified service restrictions are (20) through (21). Each route length must fall within the bounds of constraint (20). The number of stations on each route is constrained by constraint (21), as travelers prefer fewer intermediate stops between the starting and final locations, for a better experience.

## 4. Solution Algorithm

The above-proposed formulation of the mixed-integer non-linear problem is NP-hard and impractical to answer in polynomial time. Additionally, because real-trip data sources typically contain billions of travel records citywide, such as taxi-trajectory data and public-transportation transaction records, the computing burden could be extremely high. As a result, we create a sequential-heuristic approach to accomplish the goal piece by piece. The flexible-bus-routing structure is shown in Figure 1 as having two phases.

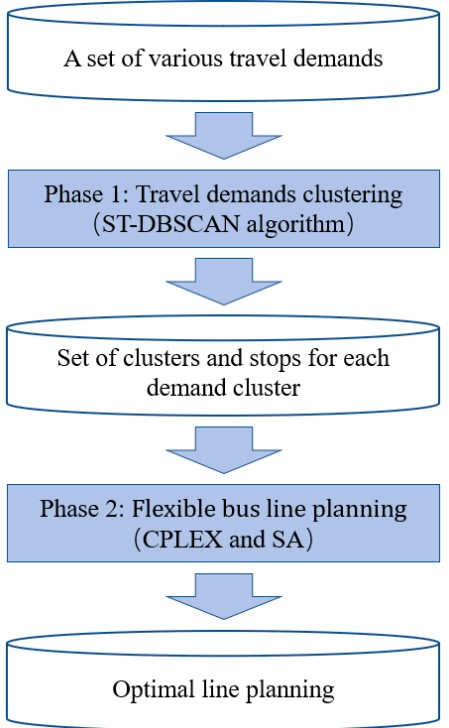

**Figure 1.** Framework of FB-routing system.

Phase 1 (Travel-demand clustering): in this phase, we identify travel patterns by grouping travel demands into groups with similar departure times and close origin and destination locations. Potential clients for flexible- bus systems are the identified demand clusters.

Phase 2 (Flexible-bus-line planning): based on the preceding mathematical model and the simulated-annealing heuristic method, a suitable flexible-bus line is constructed for each cluster with deployed flexible bus stops, which maximizes the estimated profit.

### 4.1. Travel-Demand Clustering

The ST-DBSCAN (spatial-temporary DBSCAN) algorithm was proposed by Derya et al. [54] for marine environmental research. The main drawback of the DBSCAN technique, which is addressed by ST-DBSCAN as an extension, is the inability to cluster data based on both spatial and non-spatial properties (temporal). For clustering travel demands with a temporal factor, the ST-DBSCAN technique is hence better suited. The following steps are used to implement the ST-DBSCAN algorithm:

The ST-DBSCAN (spatial-temporary DBSCAN) algorithm shown in Algorithm 1: first read the data set of objects and screen the data. Then disrupt the screened data and start the cluster algorithm. Select an object without classification and calculate the spatial distance and temporal distance between each object. Third, select a cluster of the same data point according to the distance-constraint conditions and minimum number of objects 'Minpt', and mark it. Continue the third step of the data objects in the set to expand the clusters until the object does not meet the distance constraints and the minimum number of objects

constraints. Finally, deploy stops to cover these origin locations and destination locations for each demand cluster as the importation of line planning.

---

**Algorithm 1** ST-DBSCAN Procedure

---

- Input: set of objects $D = \{d_1, d_2, d_3, \cdots, d_n\}$; Eps1: the value of temporal distance; Eps2: the value of spatial distance; Minpt: minimum number of points within Eps1 and Eps2 distance; cluster_label = 0
- Output: set of clusters $C = \{C_1, C_2, \ldots, C_k\}$

1. For $i$ = 1 to $n$
2. If $d_i$ is not in a cluster then
3. $X$ = points meeting Eps1 and Eps2 distance constraints.
4. If $|X| <$ Minpt
5. Mark $d_i$ as noise (the selected point has not enough neighbors to form a cluster).
6. Else
7. Cluster_Label = Cluster_Label + 1.
8. Mark all objects in $X$ with current cluster_label
9. Delete objects in $X$ from $D$
10. While $D$ is not empty
11. $Y$ = points meeting Eps1 and Eps2 distance constraints.
12. If $|Y| >$ Minpt
13. For $j$ = 1 to $|Y|$
14. If $d_j$ is not marked as noise and is not in a cluster
15. $d_j$ is placed into the current cluster
16. End If
17. End For
18. End If
19. End While
20. End If
21. End If
22. End For

- End

---

*4.2. Flexible-Bus-Line Planning*

Simulated annealing (SA), a well-researched local-search metaheuristic, has been widely used to solve complicated optimization problems [55]. The algorithm takes its name from metallurgical annealing, a process that involves heating and carefully controlling the cooling of a material to change its physical properties. Both of these are properties of the material that are influenced by its thermodynamic free energy. The thermodynamic free energy, also known as Gibbs energy, is impacted by heating and cooling the material. The main advantage of simulated annealing is that it allows for the escape of local optima by allowing hill-climbing actions (moves that deteriorate the objective-function value) within the goal of attaining a global optimum. Simulated annealing can be used to solve extremely challenging computational-optimization problems when exact algorithms fail; even though it typically finds an approximation of the global minimum, this may be sufficient for many real-world issues.

4.2.1. Strategic Oscillation for Dealing with Infeasible Solutions

The strategic oscillation proposed by Glover et al. [56] is utilized to permit infeasible solutions that violate tour time and vehicle-volume-range limitations. Many studies have used the strategy of "strategic oscillation," which is based on the notion of accepting impossible solution spaces in the hopes of discovering a better possible solution in subsequent cycles [57,58]. It concedes the impossibility of the solution by punishing the value of its objective function as follows:

$$Z_1(S) = Z(S) + \beta Q(S) + \gamma D(S) \tag{22}$$

$Z(S)$ refers to the value of objective Function (1) associated with solution $S$ and

$$Q(S) = \sum_{i \in S} \left[ p_i^k - Cap \right]^+ \qquad \forall k \in K \tag{23}$$

$$D(S) = \sum_{i \in V} \left[ \left( ArrT_i - T_i^k \right) x_{ij}^k \right]^+ + \sum_{i \in V} \left[ \left( T_i^k - DepT_i \right) x_{ij}^k \right]^+ \forall j \in \Delta_i^+, k \in K \tag{24}$$

where $Q(S)$ and $D(S)$ indicate the violation in vehicle capacity and time window, respectively, which correspond to constraint 10 and 14 in the mathematical model established above. $\beta$ and $\gamma$ are positive penalty coefficients corresponding to infeasibilities, and $[\cdot]^+ = \max\{0, \cdot\}$. We will set a high penalty value to avoid these key constraints for feasible solutions.

4.2.2. Simulated-Annealing Process

The simulation can be carried out using either the kinetic equations for density functions [59] or the stochastic-sampling approach [55,59,60]. The approach is an extension of the Metropolis–Hastings algorithm, a Monte Carlo technique to produce sample thermodynamic-system states, which was introduced by N. Metropolis et al. [61] in 1953.

The so-called Metropolis procedure chooses iterative stages to prevent the algorithm being trapped in a local optimum, which will lead to the best possible outcome. At the specified temperature, this procedure is applied to the simulation of atomic equilibrium. The atom is lifted slightly by a probabilistic shift $(x_i + \varsigma)$ in each phase of the process, and the change, $\Delta E$, in system energy is measured [62].

- If $E < 0$, then the movement is approved and the configuration with the modified atomic states is used as the starting state in the following operation;
- If $E > 0$, then the probability, when a new state is accepted, is:

$$P(\Delta E) = e^{\frac{-\Delta E}{kT}} \tag{25}$$

where $k$ is the Boltzmann constant, and $T$ is the temperature parameter.

By defining the system's states with $\{x_i\}$ and using the energy system as the target function, it is clear that the Metropolis technique provides a sequence of states for the specified optimization issue at a certain temperature.

Four parameters are used by SA: $T_{init}$, $T_F$, $\rho$ and $S_{max}$. $T_{init}$ denotes the beginning temperature, while $T_F$ represents the final temperature, indicating when the SA procedure is terminated. $\rho$ denotes the rate of temperature drop. $S_{max}$ stands for the largest step the algorithm can take when controlling the linear-cooling schedule.

The parameters are first established by the algorithm, which then generates an initial solution, $x_0$, at random. The best objective function is indicated by $f(x_0)$. A new solution is produced based on the existing one for each iteration. Both solutions' objective functions are assessed. The new answer will take the place of the previous best solution if $f(x_0 + \Delta x) < f(x_0)$. Otherwise, if the probability $r > exp(\Delta f / T)$, the new solution will take the place of the existing best solution. Based on linear cooling, the present temperature, $T_k$ ($T_{k+1} = \rho T_k$), is lowered. If the temperature $T_k$ is equal to $T_0$, the procedure will stop. To illustrate the direction of the suggested method, we include the SA pseudocode shown in Algorithm 2.

---

**Algorithm 2** Simulated-Annealing Algorithm

---

- Input: target nodes, initial temperature, $T_{init}$, target temperature, $T_F$, rate of temperature drop, $\rho$, maximum step, $S_{max}$.
- Output: optimal target-visiting sequence

  1.    While $T_k > T_F$ do
  2.    For number of new solution do
  3.    select a new solution: $x_0 + \Delta x$
  4.    If $f(x_0 + \Delta x) < f(x_0)$ then
  $$f_{new} = f(x_0 + \Delta x); \ x_0 = x_0 + \Delta x$$
  5.    Else
  6.    $\Delta f = f(x_0 + \Delta x) - f(x_0)$
  7.    random $r(0,1)$
  8.    If $r > exp(\Delta f / T)$ then
  9.    $f_{new} = f(x_0 + \Delta x); \ x_0 = x_0 + \Delta x$
  10.   Else
  11.   $f_{new} = f(x_0)$
  12.   End If
  13.   End If
  14.   End For
  15.   $f = f_{new}; k = k + 1$
  16.   Decrease temperature $T_k$ based on linear-cooling schedule $(T_{k+1} = \rho T_k)$
  17.   End While

- End

---

## 5. Numerical Experiments

This section provides a series of numerical experiments to illustrate the FB model constructed in Section 3 and the SA algorithm developed in Section 4. The commercial-optimization-solver CPLEX is also used to solve the MIP problem in this paper, called YALMIP, a free toolbox in MATLAB R2016a. To compare the solution efficiency of the developed SA algorithm with CPLEX, the fourth part of the objective Function (1) and the constraint (18) of the vehicle arriving time simultaneously are removed, and the model with only linear constraints is tested, based on the following small-instances special set. The computing device used in this study is a personal computer with Intel (R) Core (TM) I7-11700U 3.00 GHz CPU and 16.00 GB·RAM, using the Microsoft Windows 10 (64-bit) operating system.

### 5.1. Experiment Setting

In order to assess the model and the heuristics performance, we built situations at random and put them to the test. Three scales were used to categorize the networks: small (8 and 10 nodes), medium (16, 18 and 24 nodes), and large (more than 24 nodes) (e.g., 32, 40 and 48 nodes). The calculation was stopped, and the current answer was found to be workable but not ideal if the computational time for the mathematical model exceeded 3600 s.

In our paper, repeated test trials were run in order to forecast the best values for the related other parameters. The values for each parameter related to the SA algorithm are shown in Table 2. The SA parameters, which are presented in Table 2, include the maximum number of iterations, swap and reversion probabilities, initial annealing temperature, and rate of temperature change. The efficiency of the simulation analysis's search for the best solution and the stability of the results it produced were used to determine the parameter values. As can be seen in Table 2, as the network size grows, so do the population size and the maximum number of iterations. In general, the probabilities of swap and reversion are 0.2 and 0.5, respectively. The initial annealing temperature is 100, and the rate of temperature change is 0.99.

**Table 2.** Parameter setting for SA.

| Parameters | SA | | |
|---|---|---|---|
| | S * | M * | L * |
| Maximum iteration | 1000 | 1500 | 2000 |
| Swap probability | 0.2 | 0.2 | 0.2 |
| Reversion probability | 0.5 | 0.5 | 0.5 |
| Initial annealing temperature (°C) | 100 | 100 | 100 |
| Rate of temperature change | 0.99 | 0.99 | 0.99 |

*: S: Small-scale network; M: Medium-scale network; L: Large-scale network.

### 5.2. Experiment Results

For each experiment, ten simulation runs were carried out in order to reduce the randomness of the outcomes. It is important to note that 3600 s have been specified as the maximum computation time. The terms "$\Delta_{SA}$", which are expressed as Equation (26), present the differences between the minimal costs, labeled as $Z_S$ with SA, and those with CPLEX, denoted as $Z_C$. The final column, "$T$ ($s$)," indicates the difference between the CPU time of the CPLEX solver and the simulated-annealing algorithm. Thus,

$$\Delta_{SA} = \frac{Z_S - Z_C}{Z_C} \tag{26}$$

$$\Delta t = \frac{T_S}{T_C} \tag{27}$$

Table 3 provides a summary of the statistical data, and Figure 2 shows the average computation times using CPLEX and SA. CPLEX may be able to optimize flexible-bus networks, according to Table 4. (e.g., 8 and 10 nodes). The computing time exponentially increased with the number of nodes, making it impossible to arrive at the solution in time (e.g., 16 and 18 nodes). SA was able to obtain the same optimal solution in the small-scale instance (e.g., 8 nodes), and SA fared better in terms of faster computation times on large scales, and was able to identify solutions within the time frame.

**Table 3.** Statistical results from CPLEX and SA (10 runs).

| Nodes | CPLEX | | SA | | | | | $\Delta_{SA}$ (%) | $\Delta t$ (%) |
|---|---|---|---|---|---|---|---|---|---|
| | Cost (CNY *) | T (s) | Min * (CNY) | Mean * (CNY) | SD * (CNY) | CV * (%) | T * (s) | | |
| 8 | 2140 | 9.13 | 2140 | 2491 | 378 | 15.19 | 45.10 | 0 | 494 |
| 10 | 3105 | 474.13 | 3191 | 3665 | 445 | 12.15 | 55.75 | 2.77 | 12 |
| 16 | 4998 * | 3619.45 | 5254 | 5520 | 327 | 5.92 | 67.78 | - | - |
| 18 | 8436.5 * | 3625.27 | 8689 | 9140 | 434 | 4.75 | 75.72 | - | - |
| 24 | - | - | 23,231 | 24,014 | 643 | 2.68 | 89.36 | - | - |
| 32 | - | - | 33,695 | 36,927 | 2730 | 7.39 | 90.70 | - | - |
| 40 | - | - | 39,192 | 43,214 | 3414 | 7.90 | 119.34 | - | - |
| 48 | - | - | 46,638 | 51,407 | 2855 | 5.55 | 136.53 | - | - |

*: CNY: yuan (RMB); Min: minimized total cost; Mean: average total cost; SD: standard deviation to the mean; CV: coefficient of variation; T: average CPU time; cost (CNY): a feasible solution found at computation time 3.600 s; -: no feasible solution found.

**Table 4.** Optimized results for various scenarios.

| Results | FB with Ad Hoc Service | FB | Taxi | Gap1 (%) | Gap2 (%) |
|---|---|---|---|---|---|
| Cos.(CNY) | 23,552 | 26,345 | 47,759 | 11.86 | 102.78 |
| Veh.(veh) | 4 | 4 | 37 | 0 | 825.00 |
| Ser.(person) | 35 | 37 | 37 | 5.71 | 5.71 |
| Uns.(person) | 2 | 0 | 0 | −100.00 | −100.00 |
| Tim.(h) | 37.69 | 39.73 | 46.80 | 5.42 | 24.17 |
| Leg.(km) | 503.63 | 847.40 | 4108.93 | 68.26 | 715.86 |

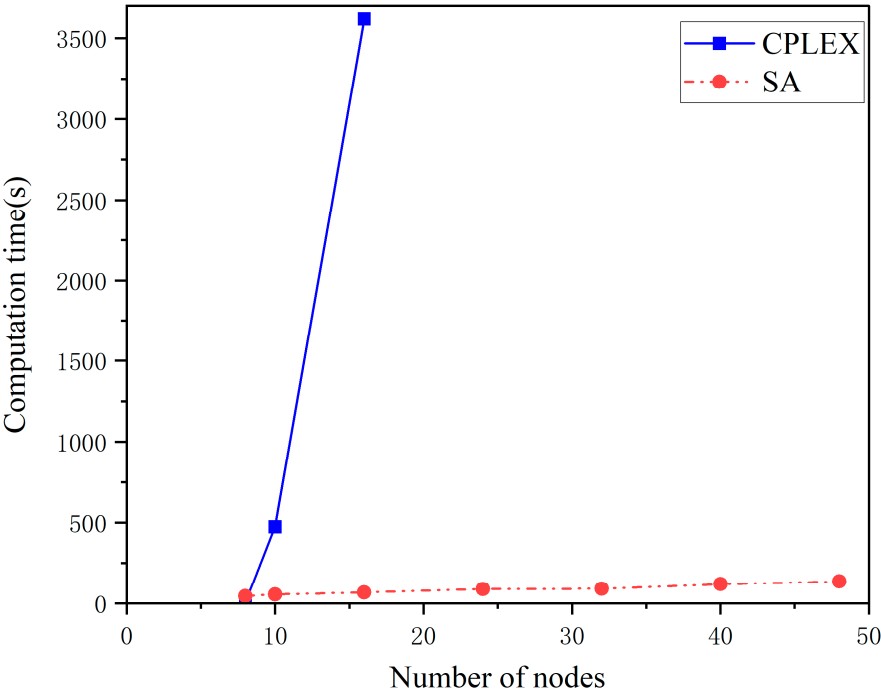

**Figure 2.** Average computation times with CPLEX and SA.

Figure 3 provides graphic representations of the convergence of the solutions for the problem instance with 18 nodes. For the identical problem instance, the graph depicts the convergence of the SA, PSO, and GA algorithms. It is obvious from the figure that PSO and GA become stuck in local optima, since it converges early for all problem situations. Because SA extensively searches the search space with the aid of an acceptance-probability strategy to allow subpar solutions, it takes more iterations to settle to a near-optimal solution. Therefore, the developed SA algorithm is effective in solving the model proposed in this paper.

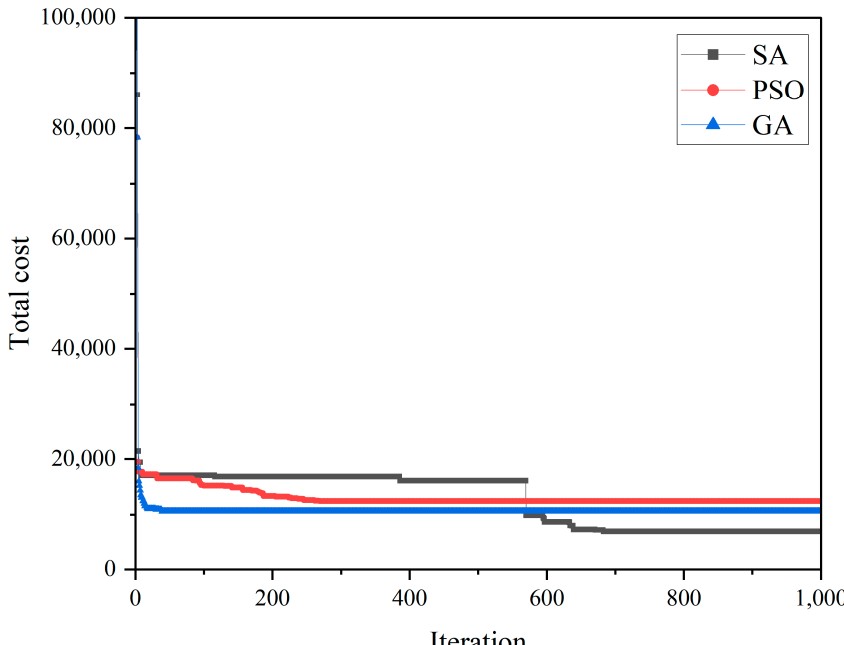

**Figure 3.** Convergence of the SA, PSO, and GA algorithms.

### 5.3. Simulations of Three Scenarios

In this section, we examine the various features of the findings for the partial-service and total-service scenarios. Since the first scenario here fits the suggested FB model, the passengers can be served by buses or an ad-hoc service. The second scenario is comparable to the FB system without ad-hoc service, as commuter buses are designed to accommodate all customers who reserve seats in advance, and guarantee that each user reaches their destination. The introduced model is mostly left identical, but we adjust the objective function and constraint (8) to represent the scenario of total service. In the second scenario, to ensure that the number of passengers served from the origin to the appropriate destination is equal to the total passengers of this *OD*, the ad-hoc cost is removed from the objective function and constraint (8) is changed, as in the following sentence:

$$\sum_{k \in K} y_{rs}^k = q_{rs}, \quad \forall r \in O, s \in D \tag{28}$$

Specifically, we used the third scenario, for the total taxi order, as the comparison group.

The experimental scenario has a simple network with 18 vertices (nine origins and their related destinations). The route between vertices is two-way. The average speed of each vehicle is set at 45 km/h, and the vehicle capacity is assumed to be 10 (unit: person). The fixed-cost per vehicle is set at CNY 500 (RMB), and the operating cost per km is CNY 18. The unserved passenger will be transported as punishment in a taxi that travels at an average speed of 60 km/h, with a fixed cost of 200 CNY/veh, and the operational cost of 8 CNY/km. The other relevant parameter values are listed below: $s_r$ = 4 CNY/min, $I_{min}$ = 20 km , $I_{max}$ = 100 km , $M$ = 10 stations. The proportions of various expenses should be taken into account while computing the objective function. Additionally, each vehicle must leave and return to the depot within the stipulated period of time, as well as visit the passenger locations within the given time window.

#### 5.3.1. Comparison of Three Scenarios

Figures 4 and 5 show the optimum routes for three scenarios using the SA algorithm for this straightforward network. The dot in orange denotes the pick-up location, the dot in green denotes the delivery site, and the number along the line denotes the order of the vehicle-access points. For instance, the first orange point denotes the place for pick up, and the 10th green point denotes its site for delivery, correspondingly. As Figure 4 showed, Vehicle 1 has access points in the following order: 0-1-10-8-17-0. Table 4 contrasts the various characteristics of the outcomes. The total cost (Cos., unit), passenger count (Pass.), vehicle count (Veh.), served-passenger count (Ser.), unserved passenger count (Uns.), travel time (Tim., unit: minute), and route length (Leg., unit: km) are notable as representing the data collected. Figure 6 summarizes the results by displaying the three scenarios' minimized overall costs.

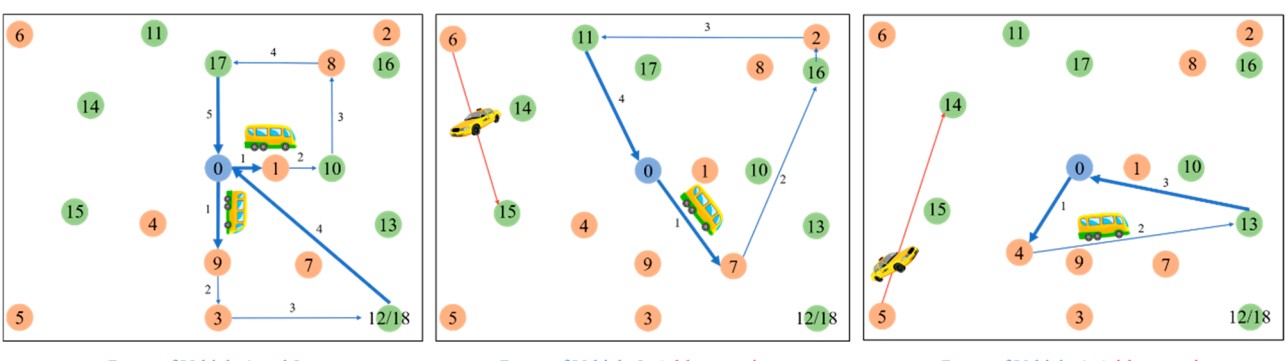

| Route of Vehicle 1 and 2 | Route of Vehicle 3; Ad-hoc service | Route of Vehicle 4; Ad-hoc service |

**Figure 4.** Results of FB with ad hoc service.

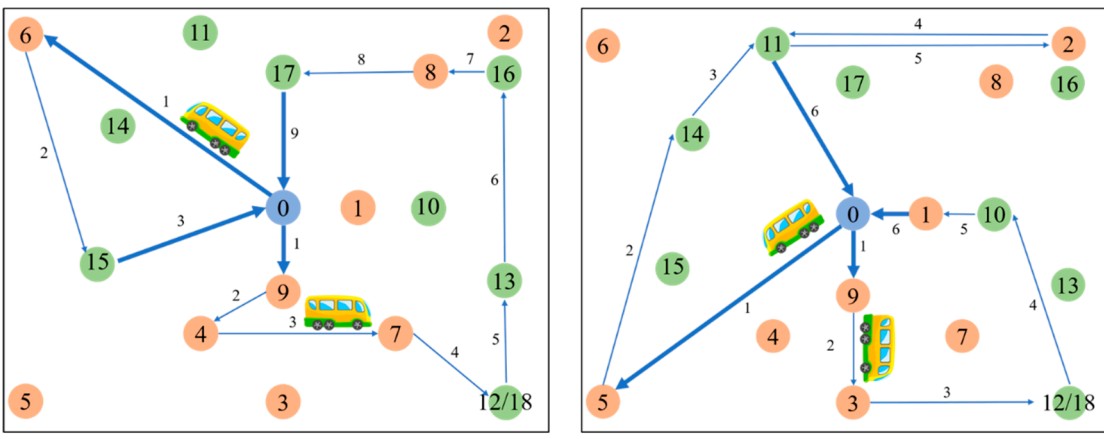

**Figure 5.** Results of FB only.

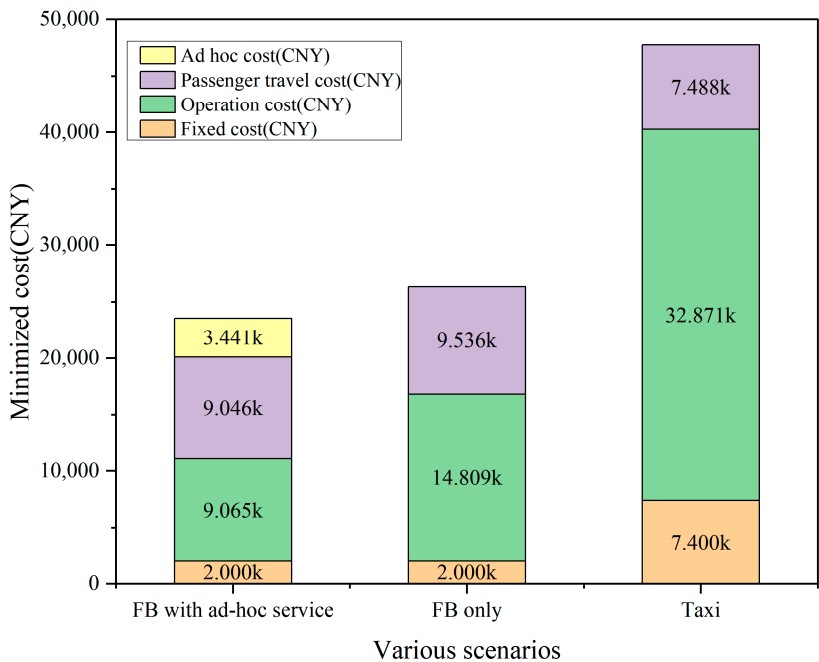

**Figure 6.** Different costs in various scenarios.

On the one hand, when comparing the outcomes of the first two scenarios, we found that the ad hoc cost of the private-cab shuttle can further lower the overall cost by 11.86% (from CNY 26,345 to CNY 23,552). From CNY 14,809 to CNY 9065, the operational cost dropped by 63.36%, and at the same time the cost of passenger transport dropped by 5.42% (from CNY 9536 to CNY 9046). On the other hand, while the cost of transportation for passengers was minimized in the taxi service, the total cost and travel lengths were significantly raised, by 102.78% and 715.86%, respectively, which would cause serious environmental issues. Overall, the findings imply that increasing the fine for using an ad hoc service operated by a private cab can reduce total costs even further, including operating costs and consumers' transportation expenses. It does not entirely adhere to the scale economics of flexible public transport, because there is an unusual demand for buses in remote locations, which will result in a longer bus route and a significant increase in passenger detour costs. Therefore, individual travelers' needs currently require the utilization of ad hoc services. By taking into account the profit between the operating firm and the passenger, the model is more accurate as regards the current transport situation.

### 5.3.2. Sensitivity Analysis of Vehicle Capacity and Speed

There is a trade-off between economies of scale of capacity and anticipated occupancy throughout the planning stage. Due to the same fixed costs, such as the driver's salary and the vehicle-registration fee, a larger vehicle has a lower operational cost per space, but seats are wasted if not enough passengers are picked up. The vehicle capacity and speed are subjected to a sensitivity analysis in this section. Four scenarios are taken into consideration, with vehicle capacities of 7, 8, 10 and 12 with operating costs of 85%, 90%, 100% and 110%, respectively. When a transportation request cannot be filled by the normal service, an ad hoc service is called in to fill the gap at the private-taxi rate outlined in Section 3.2. The speed of the vehicle is 45 km/h. The network was repeatedly solved ten times to find the optimum routing outcome according to the SA algorithm for the analyses in this part. As demonstrated in Table 5 and depicted in Figure 7, the total cost and vehicle occupancy for all scenarios are taken.

**Table 5.** Results under different vehicle capacities and operating costs.

| Results | Scenario 1 | Scenario 2 | Scenario 3 | Scenario 4 |
|---|---|---|---|---|
| Vehicle capacity | 7 | 8 | 10 | 12 |
| Factor of operating cost | 0.85 | 0.9 | 1 | 1.1 |
| Total cost (CNY) | 28,082 | 32,168 | 23,552 | 25,612 |
| Vehicle used | 2 | 3 | 4 | 3 |
| Vehicle occupancy | 59.52% | 41.67% | 48.75% | 42.59% |
| Fixed cost (CNY) | 1000 | 1500 | 2000 | 1500 |
| Vehicle-operating cost (CNY) | 5288.7 | 8589 | 9065 | 9328.5 |
| Passenger-travel-time cost (CNY) | 6440 | 6726 | 9046 | 8631 |
| Ad hoc cost (CNY) | 15,353 | 15,353 | 3441 | 6152 |

Ten-seaters are less expensive to operate than any other kind of vehicle. Due to their lower passenger-journey time cost compared to the other three situations, seven-seaters perform the best from the customers' standpoint. A smaller vehicle typically has a higher vehicle occupancy. The overall cost is greater for vehicles with capacities of seven and eight, and lowest for those with capacities of ten, which offers flexibility while reducing operational costs and ad hoc costs. Vehicles with capacities of eight and twelve have lower occupancy rates than those with capacities of seven and ten.

Vehicle speed is a significant independent variable in FB-routing systems when taking into account day and night-time traffic conditions on the route. How would changing the vehicle speed from 35 to 55 km/h affect the service patterns? Table 6 provides a summary of the findings. Figure 8 shows the total cost, vehicle occupancy, passenger-journey-time cost, and ad hoc cost.

**Table 6.** Results under different vehicle speeds.

| Results | Scenario 4 | Scenario 5 | Scenario 3 | Scenario 6 | Scenario 7 |
|---|---|---|---|---|---|
| Vehicle speed | 35 | 40 | 45 | 50 | 55 |
| Vehicle capacity | 10 | 10 | 10 | 10 | 10 |
| Total cost (CNY) | 28,161 | 25,924 | 23,552 | 23,119 | 22,134 |
| Vehicle used | 4 | 3 | 4 | 3 | 3 |
| Vehicle occupancy | 61.25% | 56.11% | 48.75% | 43.33% | 50.00% |
| Fixed cost (CNY) | 2000 | 1500 | 2000 | 1500 | 1500 |
| Vehicle-operating cost (CNY) | 8955 | 8580.6 | 9065.4 | 9972.4 | 9715.2 |
| Passenger-travel-time cost (CNY) | 11,053 | 9690.9 | 9045.6 | 8205.8 | 7477.4 |
| Ad hoc cost (CNY) | 6152.1 | 6152.1 | 3441 | 3441 | 3441 |

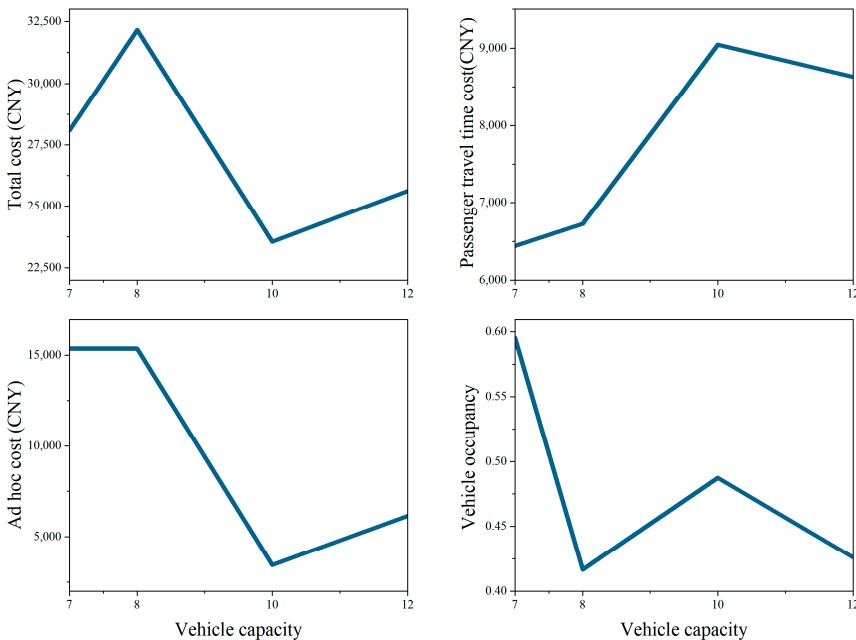

**Figure 7.** Sensitivity analysis of vehicle capacity.

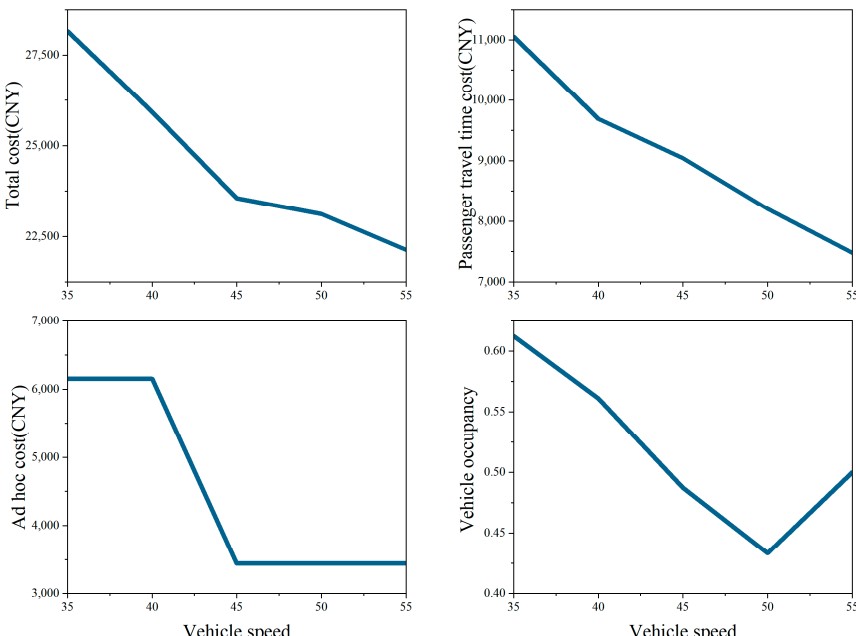

**Figure 8.** Sensitivity analysis of vehicle speed.

First, because of the shorter journey time, the overall cost falls as vehicle speed falls. As scenario seven has the maximum vehicle-speed limit, the average ad hoc cost is substantially lower than in other situations, as the ad hoc service would not be cost-effective without the speed advantage, resulting in fewer service requests being fulfilled by ad hoc services. The service quality is enhanced in terms of passenger travel time as longer-vehicle speed increases.

In conclusion, this illustration demonstrates the preference for 10-seaters. The sensitivity analysis on vehicle capacity and speed also reveals a trade-off between service quality and overall cost. Lower passenger-time costs and higher occupancy come at the expense of operation costs when capacity is reduced and speed is increased. However, there are additional considerations, such as demand characteristics, that can affect the regular service

design. In conclusion, the suggested formulation can be used to plan the deployment of vehicles with various vehicle capacities and speeds.

## 6. Case Study Based on the Shenzhen Airport Transport

The real-world transport network of Shenzhen is used in a more realistic case study to further illustrate the model's efficacy for a massive challenge [63]. Based on the reservation service function of FB, we made night-bus shuttles for passengers arriving at Shenzhen Bao'an Airport on early-morning flights.

### 6.1. Experimental Setting

The Shenzhen taxi GPS data is used to apply our algorithm to actual taxi trips. Over 43 million GPS records and 16,000 cabs per day were collected throughout the data period from February 1st to February 7th, 2019 for the Shenzhen taxi GPS data. Each GPS record contains the vehicle identification number, GPS time, longitude and latitude, passenger status, and travel speed. No personal information is present in the anonymized data. Following data pre-processing, the case study's dataset, which includes an average of 280,000 delivery trips each day, is ready. The routing probability model is fitted using the entire dataset. Additionally, the case study's data intake is decided to be the taxi data between 0 am and 3 am from February 1st.

When setting parameters for the clustering of travel demand, the unit size must be properly set when dividing the travel-demand space into units for clustering. If the size is either too large or too small, the clustering quality will suffer [64]. Based on the service radius of flexible-bus stops, which is also set at 500 m, we divide each spatial dimension of travel-demand space into intervals of 500 m [65]. The temporal component is divided into 30 min increments. Moreover, we include the detour ratios of the street networks, which is essential in the FB-routing system, to reduce the discrepancy between the Euclidean (straight-line) distance and the real distance [66]. Other settings for parameters: Table 7 provides a list of the default settings for the remaining parameters.

**Table 7.** Default parameter setting.

| Parameter | Value |
| --- | --- |
| Fixed cost per vehicle (FB) $s_{bx}$ | 500 CNY/veh |
| Fixed cost per taxi  $s_{tx}$ | 200 CNY/veh |
| Operating cost of vehicles (FB) $s_{d\_bx}$ | 18 CNY/km |
| Operating cost of taxis $s_{d\_tx}$ | 8 CNY/km |
| Travel-time cost per passenger $s_r$ | 8 CNY/min |
| Waiting-time cost per passenger $s_w$ | 4 CNY/min |
| Maximum iteration | 2000 |
| Swap probability | 0.2 |
| Reversion probability | 0.5 |
| Initial annealing temperature | 100 |
| Rate of temperature change | 0.99 |

### 6.2. Evaluation of Flexible-Bus System

6.2.1. Effectiveness of Clustering

Analysis of potential bus-reservation demand. The following restrictions were put in place to find those excessive traffic demands: the maximum walking distance (Eps1) for passengers is 2.4 km, the minimum number of passengers at the destination (Eps2) is two, and the maximum waiting time for vehicles (Minpt) is 480 sec. As a result, we are able to filter out the noise points that cannot cluster. Potential users for bus reservations are those data points that can be clustered.

As a result, the ST-DBSCAN method groups 23 sets, where the black data points are noise. Data points that share the same color belong to the same class, while those that have

distinct colors do not. There are 554 data points and 110 noise points, which together make up 19.8% of the total. The following illustration depicts the strategy, shown as Figure 9.

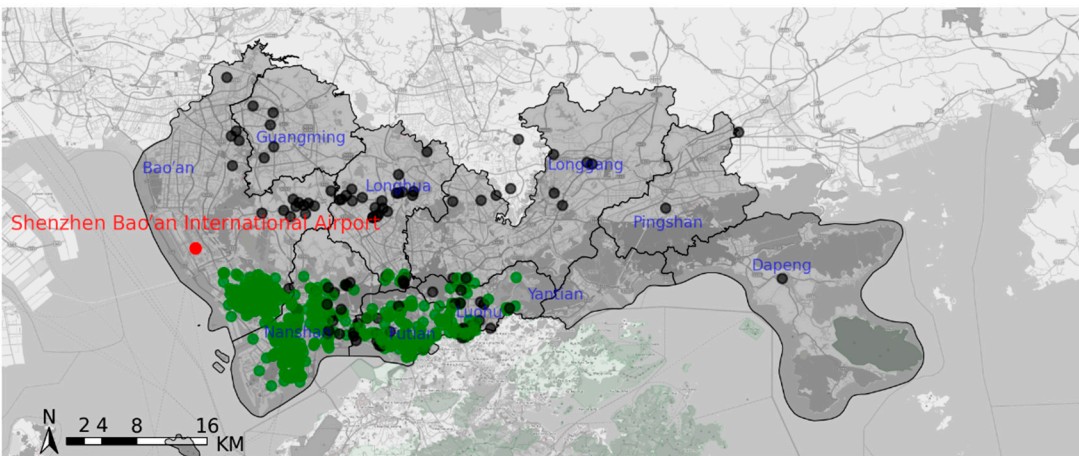

**Figure 9.** Distribution of potential demand points for FB in Shenzhen airport.

In Figure 9, the green dots represent probable passengers' destinations for bus reservations, whereas the black dots do not. The figure shows the screening of the unusual traffic demand in the outlying suburbs and the successful detection of the spatiotemporal-anomaly destination. Evidently, Bao'an District, Nanshan District, Futian District, and Luohu District have the highest potential demand for bus bookings.

Analysis of flexible-bus stations. There are 444 potential requests for bus reservations between 0 and 3 in the morning. These data are divided into bus stops using the hierarchical-clustering algorithm. The bus stop among them is designated as the center point of each category, that is, the location where the average value of longitude and latitude is found. The maximum-distance constraint is used for the distance between clusters to prevent extended walking distances once passengers get off the bus. To avoid overcrowding of public transportation, each category contains fewer than ten items, implying that the number of passengers traveling to a nearby destination in the near future is less than ten. Additionally, we assume a 40 km/h running speed for the vehicle. The following outcomes can be attained using the algorithm previously outlined.

Figure 10 shows how the hierarchical-clustering method categorizes 444 pieces of data into 131 classes. The dots of different colors in Figure 10 represent different classes, and the dots of the same color cluster into one class. Each class's number of elements is counted (see Figure 11). Naturally, there are numerous classes that have between two and five elements. This demonstrates that people can be reasonably mixed and categorized in accordance with the actual application scenario of FB public transportation.

We also calculated how far people had to walk and how long they had to wait. The first relates to the distance between the flexible-bus stop and the actual landing site obtained by taxi GPS data. The second refers to the time differential between the arrival of passengers at the beginning point and the arrival of the last passenger in the same category. Figure 12 illustrates how most travelers' walking distances are restricted to within 2 km. Additionally, the majority of passengers spend their whole waiting time inside the first 15 min. These outcomes demonstrate the efficacy of the hierarchical-clustering algorithm.

### 6.2.2. Cost Evaluation

The predicted profit balance between the bus business and the passengers is used to assess the efficiency of flexible-bus-route design. Table 8 and Figure 13 compare the outcomes of three different forms of transportation: FB with ad hoc service, FB service only, and taxi service. In addition, Table 9 shows the result of the solution FB route with ad hoc service on the Shenzhen-airport transport.

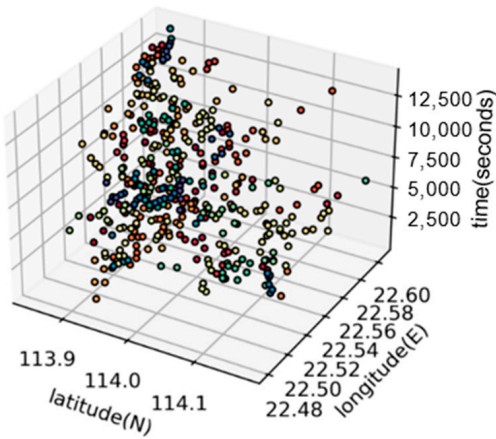

**Figure 10.** Hierarchical-clustering results.

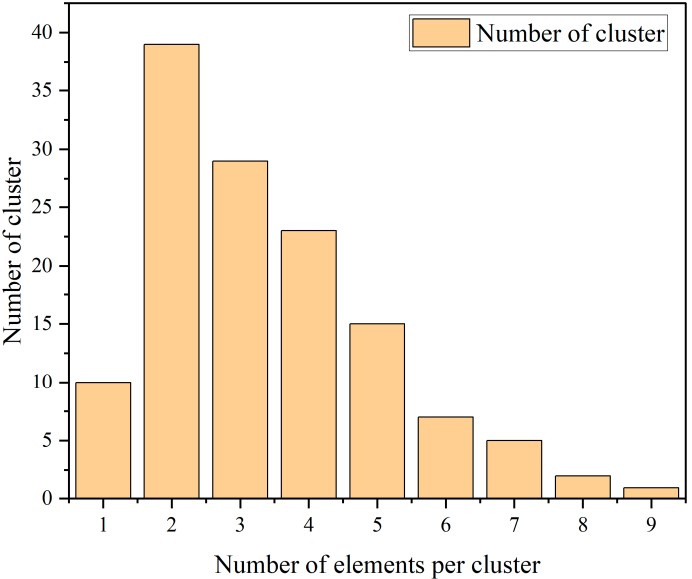

**Figure 11.** The number of elements within clusters.

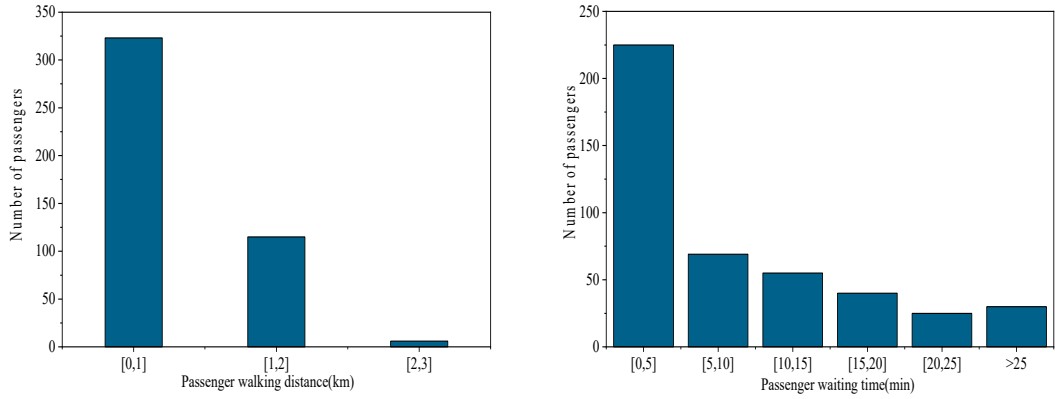

**Figure 12.** Passenger walking distance and waiting time.

**Table 8.** Optimized results for various scenarios in Shenzhen airport.

| Results | FB with Ad Hoc service | FB | Taxi | Gap 1(%) | Gap 2(%) |
|---|---|---|---|---|---|
| Cos.(CNY) | 78,301 | 84,628 | 165,410 | 8.08 | 111.25 |
| Veh.(veh) | 29 | 34 | 421 | 17.24 | 1351.72 |
| Ser.(person) | 405 | 421 | 421 | 3.95 | 3.95 |
| Uns.(person) | 16 | 0 | 0 | −100.00 | −100.00 |
| Tim.(h) | 104.62 | 124.00 | 224.96 | 18.52 | 115.03 |
| Leg.(km) | 1578.60 | 1988.70 | 9926.70 | 25.98 | 528.83 |

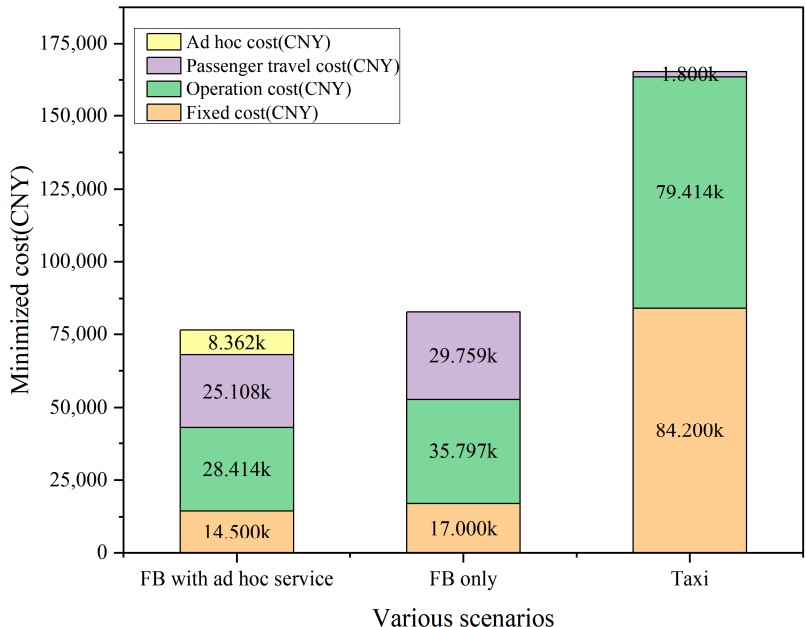

**Figure 13.** Different costs of various scenarios in Shenzhen airport.

On the one hand, we discovered that the ad hoc cost of the private-cab shuttle can further reduce the overall cost by 8.08% (from 84,628 to 78,301), when comparing the results of the first two scenarios. The operational cost decreased by 25.98%, from 35,797 to 28,414, while the cost of passenger transportation decreased by 18.52% during the same period (down to 25,108, from 29,759). However, even if the cost of passenger transportation was reduced in the third scenario (taxi service), the overall cost and journey time increased dramatically, by 111.25% and 528.83%, respectively, which would have a negative impact on the environment. Overall, the results suggest that raising the penalty for using a private cab company's shuttle service can lower total costs even more, including operating costs and consumers' transportation prices. In some remote dropping places with a single demand, the ad hoc service is preferable to the flexible bus, due to the passenger-detour time and operation profit. The model is more realistic with regard to the actual transportation situation when the profit between the operating company and the passenger is taken into account.

### 6.2.3. Sensitivity Analysis on Vehicle Capacity and Speed

Throughout the course of the planning process, there is a give-and-take between economies of scale regarding capacity and anticipated occupancy. However, if not enough people are picked up, the larger vehicle's operational cost per space will increase because of the same fixed expenditures, such as the driver's wage and the vehicle-registration charge. This section carries out a sensitivity analysis on the vehicle capacity and speed. With vehicle capacities of 15, 20, 25, and 30 and operational expenses of 90%, 100%, 110%, and 120%, respectively, four scenarios are taken into account. Ad hoc services are called in to fill in the gaps when the regular service is unable to accommodate a transportation request at

the private-taxi rate specified in Section 3.2. The bus travels at a 60 km/h speed. For the investigations in this part, the network in the case study was solved ten times in order to obtain the best possible routing result. All scenarios' total cost and vehicle occupancy are taken into account, as shown in Table 10 and in Figure 14.

**Table 9.** Result of the solution route on the Shenzhen airport transport (Total cost: CNY 78,301).

| Route of Buses (Stops Visited by the Bus) | Number of Served Passengers |
| --- | --- |
| Bus1: 0→122→103→52→12→0 | 16 |
| Bus2: 0→113→18→105→42→0 | 13 |
| Bus3: 0→125→87→128→96→0 | 15 |
| Bus4: 0→60→97→55→7→0 | 13 |
| Bus5: 0→127→32→79→8→0 | 19 |
| Bus6: 0→31→117→57→86→0 | 10 |
| Bus7: 0→126→4→29→114→0 | 17 |
| Bus8: 0→112→108→123→93→0 | 20 |
| Bus9: 0→22→63→34→16→0 | 14 |
| Bus10: 0→43→109→90→131→0 | 13 |
| Bus11: 0→2→50→104→130→0 | 9 |
| Bus12: 0→95→119→47→21→0 | 13 |
| Bus13: 0→121→24→74→118→0 | 15 |
| Bus14: 0→71→20→94→68→0 | 16 |
| Bus15: 0→56→45→44→101→0 | 13 |
| Bus16: 0→61→15→10→30→0 | 14 |
| Bus17: 0→6→129→27→89→0 | 17 |
| Bus18: 0→13→82→9→107→0 | 13 |
| Bus19: 0→51→38→14→0 | 8 |
| Bus20: 0→26→85→116→39→0 | 18 |
| Bus21: 0→46→111→106→81→0 | 16 |
| Bus22: 0→70→37→102→35→0 | 15 |
| Bus23: 0→58→75→100→92→0 | 13 |
| Bus24: 0→59→99→120→53→0 | 13 |
| Bus25: 0→115→23→69→11→0 | 16 |
| Bus26: 0→88→36→28→67→0 | 11 |
| Bus27: 0→54→98→25→17→0 | 10 |
| Bus28: 0→5→64→124→40→0 | 13 |
| Bus29: 0→19→62→91→33→0 | 12 |

More than any other type of vehicle, twenty-seaters are less expensive to operate. Thirty-seaters perform the best from the customers' perspective, since they have a reduced cost per passenger-journey time, compared to the other three scenarios. The vehicle occupancy is often higher in smaller vehicles. Vehicles with capacities of 15 and 20 have lower total costs than those with capacities of 25 and 30, which provide flexibility and higher occupancy rates while lowering ongoing operating costs and one-time charges.

Vehicle speed is a critical independent variable in the case study when considering night-time-traffic road circumstances, particularly when there is less traffic at midnight. How would changing the vehicle speed from 50 to 70 km/h affect the service patterns?

Table 11 provides a summary of the findings. Figure 15 shows the total cost, vehicle occupancy, passenger-journey-time cost, and ad hoc cost.

**Table 10.** Results under different vehicle capacities and operating costs in Shenzhen airport.

| Results | Scenario 8 | Scenario 9 | Scenario 10 | Scenario 11 |
|---|---|---|---|---|
| Vehicle capacity | 15 | 20 | 25 | 30 |
| Factor of operating cost | 0.9 | 1 | 1.1 | 1.2 |
| Total cost (CNY) | 81,137 | 78,301 | 84,951 | 86,362 |
| Vehicle used | 34 | 29 | 30 | 28 |
| Vehicle occupancy | 80.78% | 69.83% | 51.60% | 47.50% |
| Fixed cost (CNY) | 17,000 | 14,500 | 15,000 | 14,000 |
| Vehicle-operating cost (CNY) | 30,804 | 28,414 | 32,450 | 34,979 |
| Passenger-travel-time cost (CNY) | 26,704 | 25,108 | 26,306 | 24,253 |
| Passenger-waiting-time cost (CNY) | 1686 | 1917 | 1893 | 1757 |
| Ad hoc cost (CNY) | 4943 | 8362 | 9302 | 11,372 |

**Table 11.** Results under different vehicle speeds in Shenzhen airport.

| Results | Scenario 12 | Scenario 13 | Scenario 9 | Scenario 14 | Scenario 15 |
|---|---|---|---|---|---|
| Vehicle speed | 50 | 55 | 60 | 65 | 70 |
| Vehicle capacity | 20 | 20 | 20 | 20 | 20 |
| Total cost (CNY) | 85,560 | 84,479 | 78,301 | 78,308 | 77,706 |
| Vehicle used | 30 | 29 | 29 | 29 | 29 |
| Vehicle occupancy | 67.17% | 64.48% | 69.83% | 69.66% | 64.83% |
| Fixed cost (CNY) | 15,000 | 14,500 | 14,500 | 14,500 | 14,500 |
| Vehicle-operating cost (CNY) | 29,374 | 27,886 | 28,414 | 29,452 | 29,637 |
| Passenger-travel-time cost (CNY) | 29,179 | 26,917 | 25,108 | 23,683 | 22,725 |
| Passenger-waiting-time cost (CNY) | 1782 | 1779 | 1917 | 1871 | 1962 |
| Ad hoc cost (CNY) | 10,225 | 13,398 | 8362 | 8802 | 8882 |

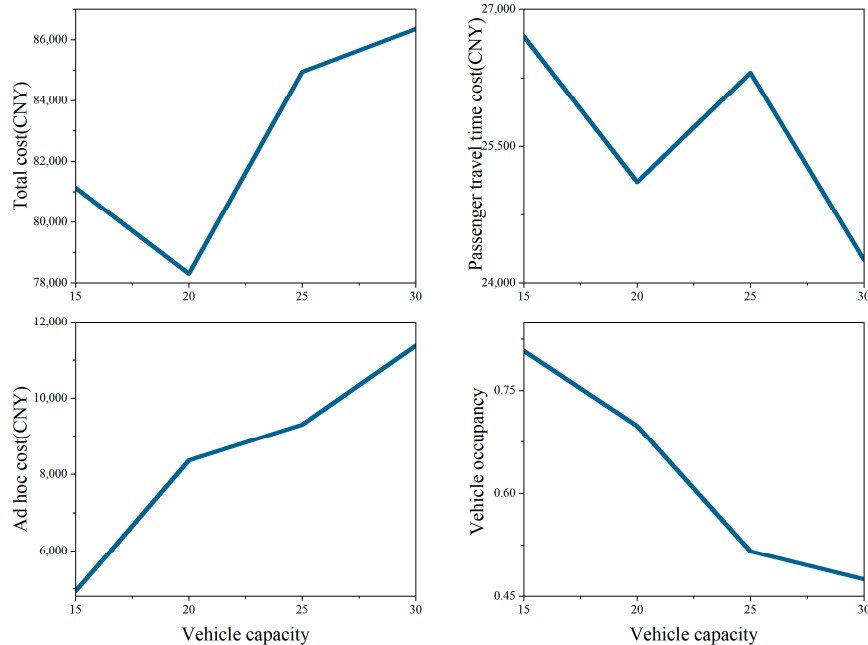

**Figure 14.** Sensitivity analysis of vehicle capacity in Shenzhen airport.

When vehicle speed increases, the overall cost decreases, due to the reduction in travel time. The passenger-travel-time cost is much lower than in other scenarios, since scenario 15 has the highest vehicle-speed limit. The lack of a clear speed advantage causes the ad

hoc service cost to remain consistent when vehicle speeds reach 60 km/h. As longer-vehicle speeds increase, the service quality is improved in terms of passenger trip time.

This example proves that 20-seaters are preferred. A trade-off between service quality and overall cost is also revealed by the sensitivity analysis of vehicle capacity and speed. When capacity is reduced and speed is enhanced, lower passenger-time costs and higher occupancy come at the expense of operation costs. However, there are still other factors to take into account, such the nature of the demand. The deployment of vehicles with a range of vehicle capacities and speeds can be planned, using the suggested formulation.

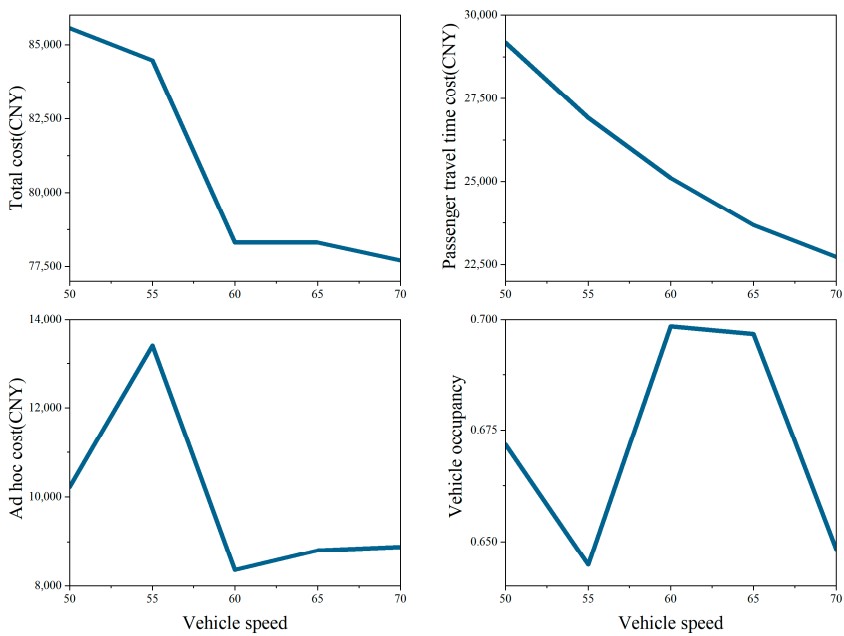

**Figure 15.** Sensitivity analysis of vehicle speed in Shenzhen airport.

This example proves that 20-seaters are preferred. A trade-off between service quality and overall cost is also revealed by the sensitivity analysis of vehicle capacity and speed. When capacity is reduced and speed is enhanced, lower passenger-time costs and higher occupancy come at the expense of operation costs. However, there are still other factors to take into account, such the nature of the demand. The deployment of vehicles with a range of vehicle capacities and speeds can be planned, using the suggested formulation.

## 7. Conclusions

In this study, we developed a flexible-bus system with an ad-hoc service to meet erratic travel demand, and we solved it with accurate and hermetic methods. Additionally, we showed that the flexible-bus system outperformed the normal flexible-bus and taxi service in terms of overall travel costs and social welfare, using actual data from Shenzhen Airport. In order to obtain the most accurate travel demand from taxi-GPS data, we specifically employed the ST-DBSCAN-clustering approach to screen out time-abnormal destinations and then choose the reserved bus stop. By explicitly including the ad-hoc service, the proposed model significantly reduces the overall cost, travel time, route length, and vehicle-vacancy rate, as compared to the practical flexible-bus system.

Future flexible-bus systems will require a systematic formulation of various pertinent practical issues, and our research will concentrate on creating the model for the following scenarios: (i) improving existing or establishing alternative-solution methods to deal with larger networks efficiently, such as applying more effective intelligence algorithms [67,68] or decomposing the problem so that the commercial solvers can directly solve it [69]; (ii) taking the time variance of each path into account, as the journey time of vehicles is strongly impacted by traffic congestion. By using hybrid-heuristic algorithms, we will also focus on

finding solutions for dynamic travel demands [70] and put more work into enhancing the quality of outcomes in terms of vehicle routes, detour time, and operation expenses.

**Author Contributions:** Conceptualization, X.C.; methodology, X.C.; software, K.R. and Y.C.; validation, X.C., K.R. and Y.C.; formal analysis, X.C.; data curation, Q.C.; writing—original draft preparation, X.C. and K.R.; writing—review and editing, Y.C.; supervision, Y.C. and Q.C.; funding acquisition, X.C. All authors have read and agreed to the published version of the manuscript.

**Funding:** This research was funded by National Key Research and Development Program of China No. 2020YFB1600400.

**Data Availability Statement:** Not applicable.

**Conflicts of Interest:** The authors declare no conflict of interest.

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
