# Peer review of "Designing Flexible-Bus System with Ad-Hoc Service Using Travel-Demand Clustering"

_mathematics, doi:10.3390/math11040825_

Round 1

Reviewer 1 Report

The article is devoted to solving an exciting problem,  some generalization "the task of a traveling salesman."

There are such notes.

1) In the expression "? ∈ ?)" see line 160. It is unclear what " ) " means in this sentence.

2) It is unclear ?_0 =  ?_{?+1} = 0 (see lines 160,161) why the service time is equal to zero in the start and finish vertexes

3) It needs the definition of set OD (see table 1).

4) It is a mistake in table 1  in definition number B, "values1e7". It should be B=10^7

5) It is unclear using the symbol of multiplying operation "dot" in formula (1)- (21). Sometimes the character is used, and sometimes the character is missed.  It should be better to choose one of the ways to present formulas. 

6) It should be better to write words instead of "s.d."  in line 177

7) It is unclear in line 200, "a fixed cost of f is generated" what does parameter f mean

8)  Notation Z'(S) is not acceptable in formula (21) line 286 because it can be  interpreted as a derivative; it would be better to use the upper line or something other

9) It needs to be clarified how the Metropolis procedure presented in 4.2.2 described the simulation of atomic equilibrium simulates the vehicle's motion. It should be a more detailed description.

10) The same notice for simulation is presented in 5.1 (lines 325- 331). It needs to be clarified what temperature simulates.

11) in table 2  (line 336), the temperature should have the scale

12) it is unclear why the rate of temperature change in table 2 (line 336) is equal to 0.99 may be better if it is equivalent to 1.00

13) there is an unclear symbol  "ï¿¥" in table 3 line 356.

14) Does ï¿¥ mean yuan in line 376?

Reviewer 2 Report

The subject of the article fits perfectly into the issues related to the sustainable development of transport, which greatly increases its value, because these issues are very important in the current times. The authors provide a good introduction to the issue, explain the existing need, clearly indicate the research gap and their contribution to science.

The literature review is appropriate, up-to-date and sufficiently extensive. Its finale is an indication of the research need, which the authors undertook in their research.

The mathematical description of the study is properly and understandably prepared, the narrative is correct and any reader will understand the assumptions and methods presented.

It is worth considering whether it is better to present the used notations at the beginning of the article or as an attachment, because the table is quite large.

The solution algorithm is also clearly explained and the use of the methods used is well justified, with an indication of computational problems.

What is important is that scientific considerations are supported by experimental application. The whole thing is correctly and properly illustrated, the figures are of good quality.

The conclusion is quite concise, it is worth emphasizing not only the practical but also the scientific nature of the study.

Some comments:

- not all literature items are prepared as required,

- it is worth adding doi numbers, which will increase the reach of cited publications,

- in my opinion, there should be a space between the word and the brackets, this applies to both recalling items in brackets, e.g.:  transportation industries[2], as well as units, e.g.: total cost(ï¿¥)

Reviewer 3 Report

Authors addressed the flexible bus system with ad-hoc service by proposing and solving a mathematical model aiming to optimise the bus stop sites, routes and schedules. The performed travel demand clustering and used simulated annealing for solving purpose. Following are my comments pertaining to the research work,

1. I want to see the motivation for using simulated annealing algorithm as there are several benchmark algorithms such as advanced PSO and GA already available in the literature. So, it is essential to highlight the limitation of those algorithms and how the simulated annealing hermetic algorithm in this paper would overcome those limitations. 

2. Authors need provide certain justification in terms of how well simulated annealing algorithms perform while aiming to balance convergence and diversity effectively. 

3. The simulate annealing hermetic algorithm has a learning process, which is used to learn from the previous experience (or iteration) and try to obtain a better solution. Could you provide a justification regarding how powerful the learning process (or how quick is the learning process) is when compared to some of the benchmark algorithms such as GA and PSO. Certain results to support the answer would be useful.

4. Thorough justification need to be provided in terms how the simulate annealing algorithm is handling the complex constraints of the mathematical model. More importantly how the algorithm will deal with infeasibility within the solution space or whether the algorithm will leave out the infeasible solution and how that process is carried out.

5. Justifications need to be strengthened to highlight how exploitation and exploration of the solution space have been dealt with by the simulated annealing algorithm. More importantly can you highlight from your computational experiments how effective is the exploitation and exploration capabilities of the algorithm when compared with other algorithms?

6. Certain justifications and comparison need to be provided with regard to some of the benchmark metaheuristics such as genetic algorithm, particle swarm optimization etc. Following papers need to cited which presents some of the state-of-the-art metaheuristic techniques.

Designing a sustainable freight transportation network with cross-docks, International Journal of Production Research

Multi-objective modelling of sustainable closed-loop supply chain network with price-sensitive demand and consumer’s incentives, Computers & Industrial Engineering

Hybridizing Basic Variable Neighborhood Search With Particle Swarm Optimization for Solving Sustainable Ship Routing and Bunker Management Problem, IEEE Transactions on Intelligent Transportation Systems

Sustainable maritime inventory routing problem with time window constraints, Engineering Applications of Artificial Intelligence

7. You have solved the mathematical model using CPLEX, although more information need to be provided in terms of the problem instances. Currently, it seems CPLEX didn't work on higher nodes such as 24, 32, 40 and 48. Have you tried decomposing the model and solving it in CPLEX to obtain optimal solution? Following research work given below can be referred to strengthen the justification and also can be used to highlight how the managerial implication need to be presented at the end which currently seems to be it weak.

Optimization model for sustainable food supply chains: An application to Norwegian salmon, Transportation Research Part E: Logistics and Transportation Review

8. More information in terms of the output deliverables of the mathematical model or values of the decision variables obtained need to be presented.

Round 2

Reviewer 3 Report

Authors have thoroughly addressed the comments and now the paper can be accepted for publication.